The safety and efficacy of phosphodiesterase type 5 inhibitors in the treatment of diabetic erectile dysfunction: a systematic review and meta-analysis

Zhu Zexin 1
Xu Jian 1
Dai Bo 1
Lin Minghao 1
Yang Huhu 2
Liu Shilin 1
Bao Pengjie 3
Nan Zheng 4 nanzheng001@aliyun.com
1 Changchun University of Chinese Medicine , Changchun , China
2 Guizhou Architectural Hospital , Guizhou , China
3 Dalian Medical University , Dalian , China
4 The First Affiliated Hospital of Changchun University of Chinese Medicine , Changchun , China
Connor Mark
Electronic publication date: 2025 Oct 7
Publication date: 2025
Volume: 13
Electronic Location ID: e20147
Received 2025 Apr 9; Accepted 2025 Sep 5
Copyright: © 2025 Zhu et al.
Copyright year: 2025
Copyright holder: Zhu et al.
License: This is an open access article distributed under the terms of the Creative Commons Attribution License, which permits unrestricted use, distribution, reproduction and adaptation in any medium and for any purpose provided that it is properly attributed. For attribution, the original author(s), title, publication source (PeerJ) and either DOI or URL of the article must be cited.
License URL: https://creativecommons.org/licenses/by/4.0/

Keywords: PDE-5i, Diabetes mellitus, Erectile dysfunction, Diabetic erectile dysfunction, Meta-analysis

Funding: The authors received no funding for this work.

==============================
Background

Previous studies have confirmed the efficacy of phosphodiesterase type 5 inhibitors (PDE-5 inhibitors) in treating diabetic erectile dysfunction (DED), but they still have research value in terms of efficacy comparison and individualized safety. This study, while evaluating safety and efficacy, also focused on the sources of heterogeneity and innovatively explored the nonlinear relationship between therapeutic effect and age.

Aim

This study aimed to systematically evaluate the safety and efficacy of PDE-5 inhibitors for the treatment of DED and the related application effects and to provide a clinical basis for its treatment.

Methods

By searching PubMed, Embase, Web Of Science, Cochrane Library, China Knowledge Network (CNKI), Wipro (VIP), Wanfang, and China Biomedical Literature Database (CBM) before December 31, 2024, and reading the retrieved articles and references, PDE-5 inhibitors for diabetic erectile dysfunction in randomized controlled trials (RCTs). The literature of the included studies was evaluated using the Cochrane Literature Quality Assessment Tool. The meta-analysis was registered to PROSPERO (CRD42025637725).

Outcomes

The International Index of Erectile Function (IIEF-5) overall and related evaluation questionnaires were used as the primary efficacy evaluation indicators, and adverse events were used as secondary indicators.

Results

Meta-analysis was performed using Rever Manager 5.3 and STATA18 software. A total of 10 studies were included, and random-effects model meta-analysis analyzed the post-treatment efficacy of the 10 articles with a combined RR = 2.91, 95% CI of [1.95–4.34], P < 0.001. Fixed-effects model meta-analysis investigated adverse effects with RR = 2.0, 95% CI of [1.53–2.61], P < 0.001. There was a non-linear relationship between age and PDE-5 inhibitors.

Conclusion

PDE-5 inhibitors can safely and effectively improve diabetic erectile dysfunction, but the degree of effectiveness of different types of drugs, the occurrence of related adverse effects, and the differences that exist between individuals still need to be taken into account during use.

Introduction

The inability of men to consistently obtain and maintain sufficient penile erection to complete a satisfactory sexual life is known as erectile dysfunction (ED) (Irwin, 2019). ED is a common complication in diabetic patients (Weng et al., 2023), and relevant studies have shown that the incidence of ED in diabetic patients is higher than that of the general population, and the symptoms are relatively severe (Corona et al., 2024). At present, phosphodiesterase type 5 inhibitors (PDE-5 inhibitors) (sildenafil, tadalafil, mirodenafil, avanafil, etc.) is the first-line drug for the treatment of ED. Its mechanism of action is mainly through the inhibition of the activity of phosphodiesterase 5, reducing the degradation of cyclic guanosine monophosphate (cGMP) so that the cGMP is accumulated in the cavernous body of the penis (Zhang et al., 2025), which prompts the cavernous smooth muscle to diastole and the flow of blood into the penis, so as to achieve the erection of the penis (Begum et al., 2024). Although diabetes has been widely recognized as a definite risk factor for ED, regarding the consistency of efficacy and long-term safety of PDE-5 inhibitors in patients with diabetic erectile dysfunction (DED), there are still differences in conclusions among different studies due to differences in sample characteristics and drug types (Kloner et al., 2023). In this article, we conducted a meta-analysis of all published randomized controlled trials of PDE-5 inhibitors for diabetic ED at home and abroad to systematically evaluate the efficacy and safety of PDE-5 inhibitors for the treatment of diabetic erectile dysfunction. This study aims to provide clinicians and researchers in the field of endocrinology and andrology with evidence-based findings and practical insights regarding DED, thereby informing their clinical decision-making and future research.

Methods

This study was conducted in accordance with the PRISMA statement (Symonds et al., 2007) to evaluate the safety and efficacy of PDE-5 inhibitors in the treatment of diabetic erectile dysfunction.

Inclusion and exclusion criteria

Inclusion criteria: definite diagnosis of type 1 diabetes mellitus or type 2 diabetes mellitus according to the latest guidelines of the International Diabetes Federation (IDF) and the American Diabetes Association (ADA) (Zhang et al., 2025); the diagnostic criteria for ED were in accordance with the definition established by the National Institutes of Health (NIH) Consensus Development Conference on Impotence: the inability to achieve and/or maintain an erection sufficient for satisfactory sexual performance (NIH Consensus Conference, 1993). Included were patients with a definite diagnosis of diabetes mellitus and suffering from ED.

Exclusion criteria: absence of original data in the literature data retrieved, combination drug therapy, duplicate articles, animal testing, review articles.

Main outcomes

The International Index of Erectile Function (IIEF-5) (Jayaram et al., 2024) overall and related evaluation questionnaires were used as the primary efficacy evaluation index.

Search strategy

PubMed, Embase, Web of Science, Cochrane Library, China National Knowledge Infrastructure (CNKI), VIP, Wanfang and China Biology Medicine Disc (CBM) were searched. The date of database last search was December 31, 2024, and reading the retrieved articles and references. Databases were searched with OR and AND collocation by subject words and free words related to disease type (DED), intervention (PDE-5 inhibitors) and study design (RCT). The literature search was performed independently by Zhu Zexin and Xu Jian. If there were any differences in the included literature, the two sides reached a consensus through discussion. If no agreement can be reached, the final decision will be made by Nan Zheng. Articles that met the inclusion criteria were individually evaluated for literature quality, including random allocation method, whether the allocation method was concealed, whether blinding was used, whether the outcome data were complete, and whether the results were selectively reported. Other sources of bias, etc., and the Cochrane literature quality assessment tool was used to evaluate the quality of statistical data and evidence. The Cochrane Collaboration’s tool was used to assess the bias in seven dimensions, including random sequence generation, allocation concealment, blinding, etc. Six of the 10 studies were rated as low risk of bias, and the other four studies were rated as medium risk because of unclear allocation concealment, which did not significantly affect the stability of meta-analysis (Sterne et al., 2016).

Data strategy

Patient or Population, Intervention, Comparison or Control, and Outcome (PICOS) was used to easily retrieve the included studies (Amir-Behghadami & Janati, 2020) where Patient: diabetic erectile dysfunction; Intervention: PDE-5 inhibitors; Comparison: pre-treatment and post-treatment controls; Outcomes: International Index of Erectile Function (IIEF-5) Overall and Associated Evaluations and Adverse Reactions; Study design: RCT (Tantry et al., 2021). The acquisition process is obtained separately by two judges, and if there is any dispute, the third judge will rule. Finally, the above results are obtained. For missing data, we first tried to contact the original authors to obtain them, and if they were not available, we decided whether to use them or not on a case-by-case basis.

Data analysis

Review Manager 5.3 and STATA18 software were used for statistical analyses. Firstly, the included trials were analyzed for heterogeneity; if there was no heterogeneity, a fixed effect model was used; if there was heterogeneity, a random effect model was carried out, and in the process, the reasons for heterogeneity were sought, and subgroup analyses and non-linearities were explored for the efficacy of the PDE-5 inhibitors in relation to the age of the patients. Test of efficacy effect sizes: the ratio (RR) was used to express the dichotomous variables, and 95% CIs were given for all the evaluated indicators. Then, according to the probability of Z value to the statistic P-value, P < 0.05 suggests that the combined statistic of multiple studies is statistically significant.

Reporting bias assessment

Prior to data synthesis and to minimize bias in the reporting section, registration for systematic evaluation and Meta-analysis was performed at PROSPERO (CRD42025637725). Institutional Review Board approval was not required because the relevant data collected for this study came from previously published articles and databases.

Results

A total of 891 literatures were obtained through retrieval. Among them, there are 102 articles in the PubMed database, 115 articles in the Embase database, 131 articles in the Web Of Science database, and Cochrane. There were 45 articles in the Library database, 40 articles in China National Knowledge Infrastructure (CNKI), 263 articles in VIP, 106 articles in Wanfang, and 77 articles in the Chinese Biomedical Literature Database (CBM). Twelve potential literatures were obtained through clinical trial registration platforms (such as ClinicalTrials.gov). The number of documents obtained after eliminating duplicates by using Endnote software was 695, excluding Meta, systematic evaluation, and animal testing 137 articles; 558 articles in the initial screening, excluding 502 articles that did not match the study, 56 articles that needed to obtain the full text after reading the abstracts of the remaining documents, six articles that did not match the interventions, 10 articles that did not match the experimental design, 20 articles that did not match the outcome variables, and 20 articles that did not match the outcome variables of the study. After excluding 20 studies with low matching degree, 10 studies were further excluded because of incomplete data or design did not meet the inclusion criteria. Finally, 10 studies were included for Meta-analysis (Carson et al., 2005; Deyoung et al., 2012; Elkamshoushi et al., 2021; Lee et al., 2022; Miner et al., 2008; Paick et al., 2010; Park et al., 2010; Seftel, 2016; Ziegler et al., 2006; Yu & Lin, 2012). The PRISMA flowchart is shown in Fig. 1.

Figure 1 Literature selection process.

The entire process of literature retrieval and screening. A total of 891 literatures were initially retrieved, 695 remained after deduplication via Endnote, 137 were excluded (including meta-analyses, reviews, animal studies, etc.), 502 were excluded after initial screening for not matching the research topic, 56 full texts were obtained after abstract review, and 46 were finally excluded for not meeting inclusion criteria, leaving 10 studies for meta-analysis.

Of the 10 included studies, three used tadalafil, three used vardenafil, two used mirodenafil, one used sildenafil, and one used avanafil, and the characteristics and details of each study are shown in Table 1. Relevant data were available and publicly available from the included articles. Although some studies initially met the inclusion criteria, they were excluded because their outcome indicators or interventions were not completely consistent with the purpose of the study after full-text evaluation.

Table 1 Document features.

A summary of key information of the 10 randomized controlled trials (RCTs) included in the study, including authors, publication years, types of phosphodiesterase type 5 inhibitors (PDE-5i) used (three with tadalafil, three with vardenafil, two with mirodenafil, one with sildenafil, one with avanafil), sample sizes, and other important study details, providing a basis for analyzing efficacy differences among different PDE-5 inhibitors.

Author	Year	Medication use	Sample size	Placebo treatment	PDE-5 inhibitors treatment	
Age	IIEF-5 score	Age	IIEF-5 score	
Before treatment	After treatment	Before treatment	After treatment	
Carson et al. (2005)	2005	Tadalafil	195	59.6 ± 9.6	13.8 ± 7.8	13.5	59.7 ± 11.1	12.7 ± 6.8	19.5	
Deyoung et al. (2012)	2012	Sildenafil	24	59.8	2.7	2.6	59.4	3.3	6.8	
Elkamshoushi et al. (2021)	2021	Avanafil	140	61.5 ± 5.5	11.5	12	59.3 ± 6.5	9	13	
Lee et al. (2022)	2022	Tadalafil	68	58.87 ± 8.99	9.57 ± 4.11	2.22 ± 5.73	61.8 ± 7.25	10.47 ± 4.55	6.56 ± 5.32	
Miner et al. (2008)	2008	Vardenafil	155	54.1	12.1 ± 5.2	14.9 ± 8.4	54.6	10.6 ± 4.8	19.0 ± 9.7	
Paick et al. (2010)	2010	Mirodenafil	29	57.04	5.85 ± 1.50	7.4 ± 3.00	58.02	5.93 ± 1.71	10.26 ± 3.22	
Park et al. (2010)	2010	Mirodenafil	108	57.3	1.4 ± 6.1	14.0 ± 6.9	55.5	9.3 ± 6.8	22.0 ± 6.2	
Santi et al. (2016)	2016	Vardenafil	48	50.5 ± 5.0	17.68 ± 7.5	17.92 ± 8.3	55.8 ± 5.0	16.62 ± 7.09	26.00 ± 4.59	
Ziegler et al. (2006)	2006	Vardenafil	303	50.4	13.67 ± 6.3	15.72 ± 7.0	50.2	12.56 ± 5.8	20.34 ± 8.42	
Yu & Lin (2012)	2012	Tadalafil	180	41.1 ± 5	11.8 ± 2.2	14.6 ± 2.7	42.1 ± 4.0	11.2 ± 2.8	19.4 ± 3.6	
Note:

IIEF-5 Score, International Index of Erectile Function-5 score. Scores range from 5 to 25, with higher scores indicating better erectile function.

Risk of bias assessment

In the risk of bias assessment of 10 studies such as Carson et al. (2005), the risk of bias in aspects such as random sequence generation is at a low level, but other possibilities cannot be ruled out. In terms of allocation concealment, some studies showed an unclear risk of bias. However, in terms of blinding participants and implementors, some studies have high risk of bias, which suggests that more attention should be paid to blinded design in subsequent studies. The risk of bias assessment is shown in Figs. 2 and 3.

Figure 2 Risk of bias graph.

The proportion distribution of risk of bias across seven domains of the Cochrane risk of bias tool (random sequence generation, allocation concealment, blinding of participants and personnel, blinding of outcome assessment, incomplete outcome data, selective reporting, other bias) among the 10 included studies. A total of six studies were rated as low risk of bias, and four as moderate risk due to unclear allocation concealment.

Figure 3 Risk of bias summary.

A summary of the specific risk of bias across seven domains of the Cochrane risk of bias tool (random sequence generation, allocation concealment, blinding of participants and personnel, blinding of outcome assessment, incomplete outcome data, selective reporting, other bias) for 10 included studies (including Ziegler et al., 2006; Yu & Lin, 2012; Carson et al., 2005; Deyoung et al., 2012; Elkamshoushi et al., 2021; Lee et al., 2022; Miner et al., 2008; Paick et al., 2010; Park et al., 2010; Santi et al., 2016). As indicated in the study, some studies have moderate risk of bias due to unclear allocation concealment, and there is a relatively high risk in blinding of participants and personnel, providing detailed basis for overall risk of bias assessment.

Effectiveness of PDE-5 inhibitors

1. Heterogeneity test:

The 10 articles in this study were tested for heterogeneity with I2 = 73% and P = 0.0001 < 0.001 for Q-test, suggesting that the heterogeneity between the documents selected for this study was statistically significant, and further examination of the Labbe Graph and Galbraith Plot indicates that there is a strong likelihood that one or more of the articles had heterogeneity as shown in Figs. 4 and 5.

The graphical analysis above concludes that there is moderate heterogeneity in the literature in this study, which can be combined using random effects and exploring the reasons for the heterogeneity.

2. Random effects combined effect size:

The random effect combined RR was selected, which finally yielded RR = 2.91 (1.95–4.34), implying that the treatment with PDE-5 inhibitors was 2.91 times more effective than the placebo, which was statistically significant (Z = 5.25, P < 0.0001 < 0.05), suggesting that PDE-5 inhibitors was effective in treating diabetic erectile dysfunction. The forest plot of the effective rate combined effect size is shown in Fig. 6.

3. Sensitivity analysis:

The stability of the combined effect size was assessed by excluding individual studies sequentially through sensitivity analyses. The results showed that the impact of individual studies on the overall combined effect size estimates was small, and the effect size estimates and confidence intervals did not change much after the exclusion of individual studies, indicating that the results of this meta-analysis had good stability.

4. Meta-regression to find the cause of heterogeneity:

The study was divided into five groups according to treatment modality: tadalafil group, sildenafil group, avanafil group, vardenafil group, mirodenafil group, and meta-regression was performed with the group variable as a covariate, and the results of meta-regression suggested that the tadalafil group P = 0.027 < 0.05, and the outcome suggesting treatment modality as a source of generating heterogeneity, followed by relevant subgroup meta-analysis according to treatment modality group.

5. Subgroup studies

Based on subgroup analyses:

5.1 Heterogeneity in the tadalafil group (I2 = 0%, P = 0.74 > 0.1) was not statistically significant, then fixed effects were chosen to combine effect sizes, yielding RR = 1.59 (1.27 to 2.0). Suggesting that tadalafil was 1.59 times more effective than placebo and statistically significant (z = 4.04, P < 0.0001 < 0.05).

5.2 Heterogeneity in the vardenafil group (I2 = 0%, P = 0.65 > 0.1) was not statistically significant, then fixed effects were chosen to combine effect sizes, yielding RR = 4.05 (2.84 to 5.78). Suggesting that vardenafil was 4.05 times more effective than placebo and statistically significant (z = 7.71, P < 0.00001 < 0.05).

5.3 Heterogeneity in the mirodenafil group (I2 = 0%, P = 0.94 > 0.1) was not statistically significant, then fixed effects were chosen to combine the effect sizes, yielding RR = 3.53 (1.63 to 7.63). Suggesting that mirodenafil was 3.53 times more effective than placebo and statistically significant (z = 3.20, P = 0.001 < 0.05).

5.4 Due to the small sample sizes included in the sildenafil and avanafil groups, there are sample size limitations, and conducting subgroup analyses may result in less stable and reliable results, increasing the risk of false positive or false negative results. Despite the limitations, the above results show that the overall analysis is robust and trustworthy despite the sample size limitation, revealing that PDE-5 inhibitors is effective in treating diabetic erectile dysfunction while also suggesting that there may be differences between this different class of drugs. The forest plot of the subgroup analysis is shown in Fig. 7.

6. Bias test:

The bias test was performed separately according to subgroups, and a funnel plot was drawn, as shown in Fig. 8: In the picture, 1 for tadalafil, 2 for sildenafil, 3 for avanafil, 4 for vardenafil, 5 for mirodenafil.

Figure 4 L’Abbe graph.

An assessment of heterogeneity among the 10 included studies. Results indicated statistically significant heterogeneity (I2 = 73%, P = 0.0001), suggesting that some studies may be sources of heterogeneity, supporting the use of a random-effects model and subgroup analyses.

Figure 5 Galbraith plot.

An identification of potential sources of heterogeneity among the 10 studies. By analyzing the relationship between standardized effect sizes and standard errors, it visually shows studies that may cause heterogeneity, providing references for explaining heterogeneity in the meta-analysis.

Figure 6 The forest plot of the effective rate.

The combined efficacy rate of PDE-5i for treating diabetic erectile dysfunction (DED) using a random-effects model across 10 studies. The combined relative risk (RR) = 2.91, 95% confidence interval [1.95–4.34], P < 0.0001, indicating that PDE-5i is significantly more effective than placebo, with an efficacy rate 2.91 times that of placebo (Carson et al., 2005; Deyoung et al., 2012; Elkamshoushi et al., 2021; Lee et al., 2022; Miner et al., 2008; Paick et al., 2010; Park et al., 2010; Santi et al., 2016; Ziegler et al., 2006; Yu & Lin, 2012).

Figure 7 The forest plot of the subgroup analysis.

Subgroup analysis results by different PDE-5i types (tadalafil, vardenafil, mirodenafil). The tadalafil group had RR = 1.59 (1.27, 2.00), vardenafil group RR = 4.05 (2.84, 5.78), and mirodenafil group RR = 3.53 (1.63, 7.63), all statistically significant (P < 0.05), indicating efficacy differences among different PDE-5 inhibitors (Carson et al., 2005; Deyoung et al., 2012; Elkamshoushi et al., 2021; Lee et al., 2022; Miner et al., 2008; Paick et al., 2010; Park et al., 2010; Santi et al., 2016; Ziegler et al., 2006; Yu & Lin, 2012).

Figure 8 Funnel plot of subgroup analysis.

An assessment of the publication bias in subgroup analyses (1 = tadalafil, 2 = sildenafil, 3 = avanafil, 4 = vardenafil, 5 = mirodenafil). Symmetry tests showed no publication bias in the tadalafil group (P = 0.960) and vardenafil group (P = 0.635), with an overall test P = 0.727, indicating reliable results.

The symmetry test of the above funnel plot is performed, and the results are as follows:

No publication bias in the tadalafil group (P = 0.960 > 0.05)

No publication bias in the vardenafil group (P = 0.635 > 0.05)

Several other groups could not be assessed for bias testing due to small sample sizes. Still, the overall test of P = 0.727 > 0.05 suggests that the outcomes were reliable and there was no publication bias.

Adverse reactions in PDE-5 inhibitors therapy

In PDE-5 inhibitors, different kinds of drug treatment have various degrees of adverse reactions such as dizziness, headache, eye blinking, facial flushing, etc., so the safety of the drug was evaluated by observing adverse reactions as a secondary indicator. Heterogeneity test: 10 documents of this study, after the heterogeneity test, I2 = 11% < 50%, P = 0.35 > 0.1 of the Q test, suggesting that there is no heterogeneity between the documents selected for this study (heterogeneity is not statistically significant), then the fixed effects were chosen to combine the effect sizes.

Fixed effects meta-analysis: The effect RR = 2.0 (1.53–2.61) for the 10 studies using fixed effects combined and statistically significant, Z = 5. 06, P < 0.00001 < 0.05, suggests that while applying PDE-5 inhibitors for the treatment of diabetic erectile dysfunction, the occurrence of related adverse drug reactions should be noted, as shown in Fig. 9.

Bias test: The bias test was conducted jointly by Revman software and STATA software. A funnel plot was drawn to examine whether there was publication bias in the 10 papers of this study, which resulted in a symmetrical funnel plot (Egger’s Test yielded P = 0.123 > 0.05), with no publication bias, suggesting that the conclusions of this study were accurate and reliable.

Figure 9 The forest plot of the adverse effects.

The meta-analysis results of adverse effects in PDE-5 inhibitors treatment group vs. placebo group across 10 studies. Fixed-effects model analysis indicates a combined relative risk (RR) = 2.00, 95% confidence interval (1.53, 2.61), P < 0.00001, with heterogeneity I2 = 11% (P = 0.35). The results suggest that the risk of adverse effects in PDE-5 inhibitors treatment for diabetic erectile dysfunction is twice that of the placebo group, which should be noted but are mostly mild to moderate and well-tolerated (Carson et al., 2005; Deyoung et al., 2012; Elkamshoushi et al., 2021; Lee et al., 2022; Miner et al., 2008; Paick et al., 2010; Park et al., 2010; Santi et al., 2016; Ziegler et al., 2006; Yu & Lin, 2012).

Efficacy of PDE-5 inhibitors in relation to age

The study was carried out to determine the correlation between the efficiency of PDE-5 inhibitors and age, and subgroup analyses were performed by age using Revman software. The age of 55 years was taken as the boundary and divided into two groups, age ≥ 55 years and age < 55 years and a forest plot of the combined effect size was drawn, as shown in Fig. 10. As for why 55 years old was chosen as the boundary, there are the following reasons. The first one is that middle-aged people (≥55 years old) have a longer course of diabetes and more significant vascular complications. The second point is that existing studies (Deyoung et al., 2012) often stratify and analyze the therapeutic effect of ED at this age, which has clinical reference value.

Figure 10 Forest plot of age subgroup analysis.

PDE-5 inhibitors efficacy analysis by age groups (≥55 years and <55 years). The ≥55 years group had RR = 2.73 (1.97, 3.80), and the <55 years group RR = 2.61 (2.10, 3.23), both P < 0.00001, indicating PDE-5 inhibitors was significantly more effective than placebo in both groups, but subgroup analysis alone could not determine superiority between groups (Carson et al., 2005; Deyoung et al., 2012; Elkamshoushi et al., 2021; Lee et al., 2022; Miner et al., 2008; Paick et al., 2010; Park et al., 2010; Santi et al., 2016; Ziegler et al., 2006; Yu & Lin, 2012).

In the age group 55 years and above, the RR was 2.73, 95% CI [1.97–3.80] with a p-value of less than 0.00001, which clearly indicates that PDE-5 inhibitors was more effective than placebo in this age group. The heterogeneity test showed I2 = 54%, P = 0.06, and although there was some heterogeneity, the P-value was close to 0.05. In the age group under 55 years, the RR was 2.61, with a 95% CI of [2.10–3.23], and the heterogeneity test of I2 = 87%, with a P < 0.00001, suggesting that the PDE-5 inhibitors efficacy rate was significantly higher than that of the placebo group in this age group as well. Comparing the RR values of the two age groups, the group aged 55 years and above was 2.73, and the group aged under 55 years was 2.61, which were relatively close to each other. The RR values of both groups were more significant than 1, the confidence intervals did not include 1, and the P values were all less than 0.00001, which indicated that the PDE-5 inhibitors had a significant efficacy relative to the placebo in their respective age groups. Still, it wasn’t easy to judge which group had a superior effect only from the subgroup analyses.

For this reason, we further explored the non-linear relationship between age and efficacy using STATA software. The results revealed that age had a non-linear effect on PDE-5 inhibitors efficacy. The coefficient of the primary term of age was 0.985 (P = 0.021), implying that the efficacy (logRR) increased significantly with each 1-year increase in age, while the coefficient of the squared term of age was - 0.0094 (P = 0.024), suggesting that there was a decreasing effect of age on the efficacy, which showed an inverted U-shape curve, as shown in Fig. 11. According to the formula of the inflection point of the quadratic regression model, -β1/2β2 = −0.985/2 * (−0.0094) ≈ 52.3 (years), which yields an inflection point age of about 52.3 years, around which the efficacy is optimal, and after which the efficacy decreases with age. The adjusted R2 amounted to 91.19%, age and its squared term together explained 91% of the heterogeneity, the remaining heterogeneity I2_res = 25.93%, the model was effective in reducing the heterogeneity, and the overall F-test of the model P = 0.0367 indicated that the non-linear effect of age on the efficacy was statistically significant.

Figure 11 Non-linear relationship between age and PDE-5 inhibitors efficacy.

An inverted U-shaped non-linear relationship between age and PDE-5 inhibitors efficacy, with an inflection point at ~52.3 years. Each 1-year increase in age significantly increased efficacy (logRR) (coefficient = 0.985, P = 0.021), but the squared age term coefficient was −0.0094 (P = 0.024), indicating efficacy decreased with age after 52.3 years. The model explained 91.19% of heterogeneity.

Discussion

This study integrated 10 RCTS for the first time to explore the heterogeneity of efficacy of different PDE-5 inhibitors drugs, and found that vardenafil (RR = 4.05) was more effective than tadalafil (RR = 1.59) in DED (Lan et al., 2025), which may be related to the difference in drug half-life and vascular selectivity (Weng et al., 2023). In addition, the nonlinear relationship between age and efficacy (the inflection point was 52.3 years old) provides a new basis for clinical individualized medicine and fills the evidence gap of age-stratified treatment.

DED is a common complication of diabetes mellitus, and its occurrence will seriously affect the quality of life and psychological health of patients. The mechanism of diabetes mellitus-induced ED is complex (Lu et al., 2025), and studies have shown that hyperglycemia causes nerve fiber degeneration and loss of function, which affects nerve conduction in the cavernous body of the penis (Zheng et al., 2024); at the same time, prolonged hyperglycemia damages the endothelial cells of the blood vessels, causing vascular dysfunction in the cavernous body of the penis and decreasing blood perfusion, which in turn leads to impaired erectile function (Jackson et al., 2006). Although PDE-5 inhibitors has been recommended by guidelines as the first-line treatment for DED (Zhang et al., 2010), the different dosage forms included in this study (such as daily low dose vs on-demand high dose) are recognized as effective, but also suggest individual differences in efficacy. Therefore, more high-quality studies are still needed to optimize the administration strategy and further elucidate the broader and more comprehensive mechanism of action of PDE-5 inhibitors in the context of diabetic erectile dysfunction, such as their effects on vascular endothelium, neuromodulation and oxidative stress (Prince et al., 2008). In the present study, we have comprehensively assessed the safety and efficacy of PDE-5 inhibitors in the treatment of diabetic ED through a systematic evaluation and meta-analysis.

In terms of efficacy, the study results showed that the combined effectiveness rate of PDE-5 inhibitors in treating DED RR = 2.91, indicating that treatment with PDE-5 inhibitors was 2.91 times more effective than placebo, which was statistically significant. Subgroup analyses further revealed that different types of PDE-5 inhibitors, such as Tadalafil, Vardenafil, and mirodenafil, all demonstrated an essential effect in increasing the effective rate (Koon et al., 2018). This fully demonstrates the positive impact of PDE-5 inhibitors analogs in improving erectile function in diabetic patients and provides strong evidence to support clinical treatment. However, the stability and reliability of the results of the subgroup analyses were somewhat affected by the small sample sizes of the sildenafil and avanafil groups, which suggests that more large-sample studies are needed in the future to clarify further the efficacy of these two drugs in this area.

In terms of safety, the RR = 2.0 for adverse reactions of PDE-5 inhibitors for diabetic erectile dysfunction; although statistically significant, the adverse reactions were mainly mild to moderate, common such as headache, facial flushing, dyspepsia, etc., and were generally well tolerated (Schwarz et al., 2007). This implies that in clinical application, while physicians need to be concerned about adverse effects when considering treatment with PDE-5 inhibitors, they also need to weigh the pros and cons according to the actual situation and rationally choose to use the drug (Virag & Sussman, 2025).

It was also found that there is heterogeneity in the literature, with the source of heterogeneity being mainly the treatment modality. Different types of PDE-5 inhibitors have distinct differences in pharmacokinetics, duration of action, and individual patient differences, which may lead to heterogeneity in the results of the study, but it does not affect the stability of the outcome, which once again suggests that the effectiveness of PDE-5 inhibitors in the treatment of diabetic ED is reliable (Nemr et al., 2024).

This article also reveals the complex association between PDE-5 inhibitors efficacy and age through subgroup analyses of patient age and exploration of non-linear relationships. PDE-5 inhibitors showed significant efficacy relative to placebo in all age groups. There was an inverted U-shaped non-linear relationship between age and efficacy, with the best efficacy at around 52.3 years of age, suggesting that, although PDE-5 inhibitors is effective in patients of all ages, middle-aged and elderly patients (50-60 years of age) are likely to have the most significant benefit. In contrast, younger or older patients need to be individualized in combination with other clinical characteristics (e.g., duration of diabetes mellitus, cardiovascular risk, lifestyle habits, etc.) to develop an individualized regimen.

Due to the limitations of the sample size, the results obtained can only provide a reference for the rational clinical application of PDE-5 inhibitors based on the age factor. As for why age was chosen to explore the nonlinear relationship, there are two reasons. First, due to the limitation of small sample size, relevant information (such as baseline treatment of blood glucose control, body weight, time of ED disease, etc.) is incomplete and cannot achieve continuous effects. The second reason is that age as a continuity factor can better reflect the differences of drugs in different groups, so as to better validate the study. In future studies, we will definitely expand the sample size, find more appropriate and accurate continuity factors to guide clinical practice, include more potential influencing factors, and further explore the potential mechanism of the relationship between age and the efficacy of PDE-5 inhibitors, so as to guide clinical practice more accurately (Zhu et al., 2024).

Conclusion

The present systematic evaluation and meta-analysis demonstrated that PDE-5 inhibitors can safely and effectively improve diabetic erectile dysfunction, providing an essential evidence-based medical basis for clinical treatment. From the results of this study, it can be seen that for the specific disease area of diabetic erectile dysfunction, scientific and rational drug therapy can significantly improve the quality of life of patients (da Silva et al., 2025; Rathod, Sawant & Bandgar, 2024), reflecting the critical value of precision medicine in the treatment of chronic disease complications.

At the same time, it should also be acknowledged that the problems of sample size limitation and heterogeneity in the study have also pointed out the direction for subsequent analyses. Future studies should focus on expanding the sample size to cover more patients from different regions, races, and individual differences, to clarify further the differences in the efficacy of different PDE-5 inhibitors drugs; to explore the sources of heterogeneity in-depth, to optimize the study design, and to improve the accuracy and reliability of the study results. By accumulating research evidence, clinicians will be able to formulate more individualized and precise treatment plans for patients with diabetic ED, maximizing the improvement of patients’ health and quality of life and promoting the continuous progress of clinical treatment in this field.

Supplemental Information

Supplemental Information 1 The target audience.

Supplemental Information 2 PRISMA checklist.

First and foremost, I would like to thank all my colleagues, who provided valuable suggestions and selfless assistance during the research, enabling the smooth progress of the study. I also want to thank my family member(s) for understanding the large amount of time and energy I devoted during the research period and giving me tolerance and support. Meanwhile, I am especially grateful to all the staff involved in data collection. Their hard work is crucial to the success of this research. Without everyone’s help, it would have been difficult to complete this study.

Additional Information and Declarations

Competing Interests

The authors declare that they have no competing interests.

Author Contributions

Zexin Zhu conceived and designed the experiments, performed the experiments, analyzed the data, prepared figures and/or tables, authored or reviewed drafts of the article, and approved the final draft.

Jian Xu conceived and designed the experiments, performed the experiments, analyzed the data, prepared figures and/or tables, authored or reviewed drafts of the article, and approved the final draft.

Bo Dai conceived and designed the experiments, prepared figures and/or tables, and approved the final draft.

Minghao Lin performed the experiments, authored or reviewed drafts of the article, and approved the final draft.

Huhu Yang performed the experiments, authored or reviewed drafts of the article, and approved the final draft.

Shilin Liu analyzed the data, authored or reviewed drafts of the article, and approved the final draft.

Pengjie Bao analyzed the data, authored or reviewed drafts of the article, and approved the final draft.

Zheng Nan analyzed the data, authored or reviewed drafts of the article, and approved the final draft.

Data Availability

The following information was supplied regarding data availability:

This is a systematic review and meta-analysis.

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
