# Peer review of "The safety and efficacy of phosphodiesterase type 5 inhibitors in the treatment of diabetic erectile dysfunction: a systematic review and meta-analysis"

_PeerJ, doi:10.7717/peerj.20147_

## Round 0.1 · original submission · Major Revisions

· Academic Editor

Major Revisions

**Language Note:** The review process has identified that the English language must be improved. PeerJ can provide language editing services - please contact us at [email protected] for pricing (be sure to provide your manuscript number and title). Alternatively, you should make your own arrangements to improve the language quality and provide details in your response letter. – PeerJ Staff

Reviewer 1 ·

Basic reporting

I read with interest this review on the efficacy of phosphodiesterase 5 inhibitors in the treatment of diabetic erectile dysfunction. Revise the manuscript to clearly define its unique contribution to the field. Although the topic is relevant and timely, the current version of the paper does not offer any novel insights or perspectives beyond what is already available in the literature. It is essential to emphasize how this review differs from existing work and to articulate the specific value it adds to the academic discourse. The manuscript is well written, and the author effectively outlines the clinical and diagnostic challenges associated with this complex and multifactorial condition. Maintain the clarity of the writing, but expand the discussion of diagnostic issues with more detailed, evidence-based analysis to enhance the paper’s clinical relevance and depth. The figures, writing style, study design, interpretations, and overall quality of the work are satisfactory. Retain these strengths, but ensure that all visual elements are clearly labeled, well-integrated into the narrative, and directly support the key arguments presented. The reference list is currently insufficient. Substantially expand it to include recent and relevant studies, ensuring that all major claims are supported by appropriate citations. A comprehensive and up-to-date literature review is necessary to establish the credibility and scholarly value of the manuscript. Ultimately, the paper in its current form does not contribute new knowledge to the field. Reframe the discussion and conclusions to highlight original insights or emerging perspectives. Without a clearly defined and novel contribution, the manuscript does not meet the standards required for publication and requires major revisions.

Experimental design

.

Validity of the findings

.

·

Basic reporting

This manuscript is prepared in English. Literature references are appropriately selected.
Article structure, tables, and raw data are correctly prepared, except that Figure 10 cannot be read due to its very low resolution.

The Introduction section is sufficient in general. However, the authors stated in Line 34 that there is controversy regarding the efficacy and safety of PDE-5 inhibitors in treating diabetic erectile dysfunction. In my opinion, there is no controversy regarding the efficacy and safety of PDE-5 inhibitors in treating diabetic erectile dysfunction. Diabetes is a well-recognized risk factor for ED. Please consider revising your statement to reflect this issue. Despite my disagreement, the authors should support their hypothesis if they still advocate their hypothesis using literature references in this regard.

Experimental design

This manuscript clearly defines the research question that might fill the gap in the literature.
The methods section is described with sufficient information to be reproducible by another study.

Validity of the findings

The manuscript clearly states its contribution to the literature, and the statistical methods are well-described. The conclusion is also clearly articulated.

Additional comments

In general, phosphodiesterase type 5 inhibitors (PDE-5i) are used as the first-line treatment for diabetes-related erectile dysfunction. However, dosing strategies may vary. Some authors advocate for a low-dose daily regimen, while others prefer high-dose on-demand use. This variation should be acknowledged and discussed.

Line 33: To maintain consistency, please use uniform terminology throughout the manuscript. Choose either 'phosphodiesterase type 5 inhibitors' or another accurate term, but note that 'Phosphate 2 esterase five inhibitors' is incorrect and should not be used.

Line 34. Please correct the term “PDE-5”.

Line 34. In my opinion, there is no controversy regarding the efficacy and safety of PDE-5 inhibitors in treating diabetic erectile dysfunction. Diabetes is a well-recognized risk factor for ED. Please consider revising your statement to reflect this issue.

Line 120. Please clarify the methodology—how were the 12 papers obtained? What channels or databases were used?

Figure 8. All tables and figures should be self-explanatory. Please include a legend or code to identify the symbols used for each PDE-5i agent.

Figure 10. The figure is difficult to read due to low resolution. Please provide a higher-quality version.
Line 207. Please explain the rationale behind using a 55-year age cutoff. Why was this specific value chosen?

Reviewer 3 ·

Basic reporting

The manuscript presents a well-structured systematic review and meta-analysis. However, some passages, particularly in the abstract and introduction, contain phrasing that could be refined for clarity and fluency. A careful language revision may help ensure that the scientific content is communicated as effectively as possible. Please revise the abstract to correct unusual phrasing (e.g., “phosphate 2 esterase five inhibitors”) and improve overall clarity.

Several figure legends are overly brief and do not provide sufficient information for interpretation without referring to the main text. In particular, the forest plots (Figures 2 and 3) and the spline curve (Figure 5) lack clarity regarding abbreviations and contextual meaning. Should the authors consider revising all figure legends to include full definitions of abbreviations, descriptions of comparison groups, and brief interpretations of the visualized data?

Experimental design

The use of the Cochrane Risk of Bias tool is mentioned, but the criteria applied, the judgments made, and their influence on the synthesis are not fully described. Further methodological transparency in this regard would enhance the credibility of the findings. Could more detail be provided on how the tool was implemented and how its outcomes informed the overall interpretation?

Although the manuscript states that the literature search was conducted up to December 2024, it does not specify the exact date the databases were last queried. Precise reporting of the search timeline is critical for reproducibility. Might the authors specify the exact date of the final search and ensure it is clearly stated in the Methods section?

A more detailed discussion of the methodological limitations of the included studies, such as small sample sizes, lack of blinding, and inconsistencies in reporting, would support a more balanced interpretation of the findings. Would it be possible to expand the limitations section to explicitly address these study-level concerns and their implications for the overall conclusions?

Validity of the findings

-

---

## Round 0.2 · Minor Revisions

· Academic Editor

Minor Revisions

The authors have revised this review to the satisfaction of the original reviewers, but with a "fresh" set of eyes, I still think there are some issues that need addressing.

Firstly, this is not the first systematic review/meta-analysis to address this topic, and previous work should be acknowledged - e.g. PMID: 26180759; PMID: 34557099; PMID: 30523399 - which all reached similar conclusions. That does not mean the present study was not worthwhile but it is important to place the results into context. I would also caution against using this single study to "provide treatment recommendations for DED" (line 70/71) ... recommendations/guidelines usually come from a consensus from an overarching professional body, not individual investigators.

There are still aspects of the manuscript that are unclear/need further clarification.

- What is the reference for the definition of ED, especially as used in the studies analyzed - at the moment all there is is a vague comment about the NIH, which is not helpful (line 79/80).

- Table 1 does not define what was measured - what are the values in the "Before Treatment" and "After Treatment" columns ? Are they IEFF-5 scores ?

- In Figures 5 & 6 & 9 it is not clear why the points all favour placebo (i.e., are on the right-hand side of the graph). This may be because the outcome plotted is not clear. If the drugs favoured successful erections/adverse effects, surely the points would be on the drug side, if placebo favoured failure to achieve erection then perhaps they would sit on the placebo side - but this would seem an odd way to present the data. Please clarify what the RR actually refers to, I may be missing the point here

- Can the authors please provide a simple explanation of what they are measuring when the talk about heterogeneity ? I know what the word means, but precisely what were you putting a number on ? I can guess what the different sized circles in Figure 4 represent (sample size ?), but it would be nice to know for sure.

- line 131, lines 135-137 , lines 141-142 do not really make sense - I am not sure what the authors are trying to communicate.

- line 232 - what does "according to the public notice" mean in this context ?

- line 252; given that (by definition) the drugs studied here are PDE5 inhibitors, it is unclear what the authors mean by "confirm a fuller and broader mechanism of action"

·

Basic reporting

Acceptable English writing throughout the manuscript

Sufficient literature is provided.

Professional article structure, figures, and tables. Raw data shared.

Self-contained with relevant results to hypotheses.

Experimental design

Methods are described with sufficient detail and information to replicate.

Validity of the findings

All underlying data have been provided; they are robust, statistically sound, and controlled.

Additional comments

Thank you for your prompt response to my previous comments.

Reviewer 3 ·

Basic reporting

-

Experimental design

-

Validity of the findings

-

Additional comments

The authors have demonstrated careful attention to the revisions, addressing the points raised in the previous review both adequately and thoughtfully. Notable improvements were made in methodological transparency, including clearer explanations regarding the use of the risk of bias assessment tool, precise reporting of the literature search timeline, and a more detailed discussion of the methodological limitations of the included studies. The figure legends have also been revised to become more comprehensive and self-explanatory, and the manuscript has undergone linguistic editing, resulting in improved clarity and overall readability.

In light of the revisions made and the relevance of the topic, the manuscript is well-prepared for publication.

---

## Round 0.3 · accepted · Accept

· Academic Editor

Accept

The authors have satisfactorily addressed my remaining concerns about the manuscript. It is good to go.